# VIDEO SUPER-RESOLUTION TRANSFORMER WITH MASKED INTER&INTRA-FRAME ATTENTION

## ABSTRACT

Recently, Vision Transformer has achieved great success in recovering missing details in low-resolution sequences, i.e. the video super-resolution (VSR) task. Despite its superiority VSR accuracy, the heavy computational burden as well as the large memory footprint hinders the deployment of Transformer-based VSR models on constrained devices, e.g. smart phones and consumer electronic products. In this paper, we address the above issue by proposing a novel feature-level masked processing framework: VSR with **M**asked **I**ntra and inter frame **A**ttention (MIA-VSR). The core of MIA-VSR is leveraging feature-level temporal continuity between adjacent frames to reduce redundant computations and make more rational use of previously enhanced SR features. Concretely, we propose an intra-frame and inter-frame attention block which takes the respective roles of past features and input features into consideration and only exploits previously enhanced features to provide supplementary information. In addition, an adaptive block-wise mask predicting module is developed to skip unimportant computations according to feature similarity between adjacent frames. We conduct detailed ablation studies to validate our contributions and compare the proposed method with recent state-of-the-art VSR approaches. The experimental results demonstrate that MIA-VSR improves the memory and computation efficiency over state-of-the-art methods, without trading off PSNR accuracy.

## 1 INTRODUCTION

Image super-resolution (SR) refers to the process of recovering sharp details in high resolution (HR) images from low resolution (LR) observations. Due to its great value in practical usages, e.g., surveillance and high definition display, SR has been a thriving research topic over the last twenty years. Generally, compared with single image super-resolution which only exploits intra-frame information to estimate the missing details, video super-resolution (VSR) additionally leverages the temporal information to recover the HR frames and therefore often leads to better SR results.

The key of VSR lies in making rational use of temporal information. Researches in the early stage utilized convolutional neural networks (CNNs) to extract features and have investigated advanced information propagation strategies (Fuoli et al., 2019; Isobe et al., 2020a; Chan et al., 2021; 2022; Shi et al., 2022), sophisticated alignment modules (Wang et al., 2019; Chan et al., 2021; Tian et al., 2020; Liang et al., 2022b; Shi et al., 2022; Xu et al., 2023), effective training strategies (Xiao et al., 2021) as well as elaborately designed network architectures (Li et al., 2020; Isobe et al., 2022) for the pursuit of highly accurate VSR results. Recently, with the rapid development of Transformers in computer vision, several attempts have been made to exploit Transformers for better recovering missing details in LR sequences. VRT (Liang et al., 2022a) proposed a parallel VSR Transformer with deformable convolution to model long-range temporal dependency. RVRT (Liang et al., 2022b) introduced a recurrent structure with guided deformable attention alignment in the VSR Transformer of the parallel structure. Shi et al. (2022) proposed a recurrent VSR Transformer with bidirectional propagation and patch alignment. Due to the powerful representation learning capabilities of self-attention, these Transformer-based approaches have raised the state-of-the-art in VSR to a new level.

In spite of its superior SR results, the heavy computational burden and large memory footprint (Liang et al., 2022a;b; Shi et al., 2022) of Transformer-based VSR approaches limits their application in constrained devices. With the common availability of video cameras, efficient video processing have

became an increasingly important research topic in the literature of computer vision. Paralleling to network pruning and quantization, which is applicable to all kinds of applications, exploiting temporal redundancy for avoiding unnecessary computation is a specific strategy for accelerating video processing. In their inspiring work, Habibian et al. (2021) proposed a skip-convolution method which restricts the computation of video processing only to the regions with significant changes while skipping the others. Although the skip-covolution strategy could save computations without significant performance drop in high-level tasks, e.g. object detection and human pose estimation, as low-level vision tasks such as video super-resolution are highly sensitive to minor changes in image content, whether such a skip processing mechanism is applicable to VSR is still an open question. In this paper, we provide an affirmative answer to the above question with a novel masked VSR framework, i.e. **M**asked **I**nter&Intra frame **A**ttention (MIA) model. To the best of our knowledge, our work is the first attempt in the field of low-level video processing which explicitly makes use of temporal continuity to skip unimportant computations.

Our MIA-VSR model advances the existing VSR approaches in the following two aspects. Firstly, we develop a tailored inter-frame and intra-frame attention block (IIA) for making more rational use of previously enhanced features. Instead of directly inheriting Swin-Transformer block (Liu et al., 2021) to process concatenated hidden states and image feature, our proposed IIA block takes the respective roles of past features and input features into consideration and only utilize image feature of the current frame to generate query token. As a result, the proposed IIA block not only reduces the computational consumption of the original joint self-attention strategy by a large margin, but also aligns the intermediate feature with spatial coordinate to enable efficient masked processing. Secondly, we propose a feature-level adaptive masked processing mechanism to reduce redundant computations according to the continuity between adjacent frames. Different from the previous efficient processing models which simply determine skipable region of the whole network according to pixel intensities, we adopt a feature-level selective skipping strategy and pass by computations of specific stage adaptively. The proposed feature-level adaptive masking strategy enables our MIA-VSR to save computations while at the same time achieve good VSR results. Our contributions can be summarized as follows:

- We present a feature-level masked processing framework for efficient VSR, which could leverage temporal continuity to reduce redundant computations in the VSR task.

- We propose an intra-frame and inter-frame attention block, which could effectively extract spatial and temporal supplementary information to enhance SR features.

- We propose an adaptive mask predicting module, which masks out unimportant regions according to feature similarity between adjacent frames for different stages of processing.

- We compare our MIA-VSR model with state-of-the-art VSR models, against which our approach could generate superior results with less computations and memory footprints.

## 2 RELATED WORK

### 2.1 VIDEO SUPER-RESOLUTION

According to how the temporal information is utilized, the existing deep learning based VSR methods can be grouped into two categories: temporal sliding-window based methods and recurrent based methods.

**Temporal sliding-window based VSR.** Given an LR video sequence, one category of approaches process the LR frames in a temporal sliding-window manner which aligns adjacent LR frames to the reference frame to estimate a single HR output. The alignment module plays an essential role in temporal sliding-window based method. Earlier works (Caballero et al., 2017; Liu et al., 2017; Tao et al., 2017) explicitly estimated optical flow to align adjacent frames. Recently, implicit alignment modules (Xu et al., 2023) were proposed to perform alignment in the high dimensional feature space. Dynamic filters (Jo et al., 2018), deformable convolutions (Dai et al., 2017; Tian et al., 2020; Wang et al., 2019) and attention modules (Isobe et al., 2020b; Li et al., 2020) have been developed to conduct motion compensation implicitly in the feature space. In addition to the alignment module, another important research direction is investigating sophisticated network architectures to process the aligned images. Many works were aimed at estimating HR image from multiple input images

in a temporal-sliding window manner (Li et al., 2020; Wang et al., 2019; Cao et al., 2021; Liang et al., 2022a). Although the alignment module enables sliding window based VSR networks to better leverage temporal information from adjacent frames, accessible information to the VSR models are limited by the size of temporal sliding window and these methods could only make use of temporal information from limited number of input video frames.

**Recurrent framework based VSR.** Another category of approaches apply recurrent neural networks to exploit temporal information from more frames. FRVSR (Sajjadi et al., 2018) firstly proposed a recurrent framework that utilizes optical flow to align the previous HR estimation and the current LR input for VSR. RLSP (Fuoli et al., 2019) propagates high dimensional hidden states instead of previous HR estimation to better exploit long-term information. RSDN (Isobe et al., 2020a) further extended RLSP (Fuoli et al., 2019) by decomposing the LR frames into structure and detail layers and introduced an adaptation module to selectively use the information from hidden state. BasicVSR (Chan et al., 2020) utilized bi-directional hidden states, and BasicVSR++(Chan et al., 2022) further improved BasicVSR with second-order grid propagation and flow-guided deformable alignment. PSRT(Shi et al., 2022) adopted the bi-directional second-order grid propagation framework of BasicVSR++ and utilized multi-frame self-attention block to jointly process previous frame's feature propagation outputs and input features. Generally, by passing the high dimensional hidden states or output feature, the recurrent based VSR could incorporate temporal information from more frames for estimating the missing details and therefore achieve better VSR results. Our proposed MIA-VSR follows the general framework of bi-directional second-order hidden feature propagation while introduces masked processing strategy and intra&inter-frame attention block for the pursuit of better trade-off between VSR accuracy, computational burden and memory footprint.

## 2.2 EFFICIENT VIDEO PROCESSING

Various strategies of reducing temporal redundancy have been explored for efficient video processing. One category of approaches adopt the general network optimization strategy and utilize pruning (Xia et al., 2022) or distillation (Habibian et al., 2022) methods to train light-weight networks for efficient video processing. In order to take extra benefit from the temporal continuity of video frames, another category of methods only compute expensive backbone features on key-frames and apply concatenation methods (Jain et al., 2019), optical-flow based alignment methods (Zhu et al., 2017; Li et al., 2018; Zhu et al., 2018; Jain et al., 2019; Nie et al., 2019; Hu et al., 2020), dynamic convolution methods(Nie et al., 2019) and self-attention methods(Hu et al., 2020) to enhance features of other frames with key-frame features. Most recently, (Habibian et al., 2021) proposed a skip-convolution approach which only conducts computation in regions with significant changes between frames to achieve the goal of efficient video processing. However, most of the above advanced efficient video processing schemes were designed for high-level vision tasks such as object detection and pose estimation. In the literature of VSR, most attempts of efficient video processing were made on reducing per-frame computations with pruning (Xia et al., 2022) and distillation (Habibian et al., 2022) techniques. To the best of our knowledge, our study is the first work that leverages temporal continuity across different areas to reduce redundant computation for low-level VSR task.

## 3 METHODOLOGY

### 3.1 OVERALL ARCHITECTURE

Given $T$ frames of low-resolution video sequence $I_{LR} \in \mathbb{R}^{T \times H \times W \times 3}$, our goal is to reconstruct the corresponding HR video sequence $I_{HR} \in \mathbb{R}^{T \times sH \times sW \times 3}$, where $s$ is the scaling factor and $H, W, 3$ are the height, width and channel number of input frames.

We built our MIA-VSR framework upon the bi-directional second-order grid propagation framework of BasicVSR++ (Chan et al., 2022), which has also been adopted in recent state-of-the-art method PSRT (Shi et al., 2022). The whole model consists of three parts, i.e. the shallow feature extraction part, the recurrent feature refinement part and the feature reconstruction part. We follow previous works BasicVSR++ (Chan et al., 2022) and PSRT (Shi et al., 2022) which use plain convolution operation to extract shallow features and adopt pixel-shuffle layer (Shi et al., 2016) to reconstruct HR output with refined features. The patch alignment method used in PSRT (Shi et al., 2016) is

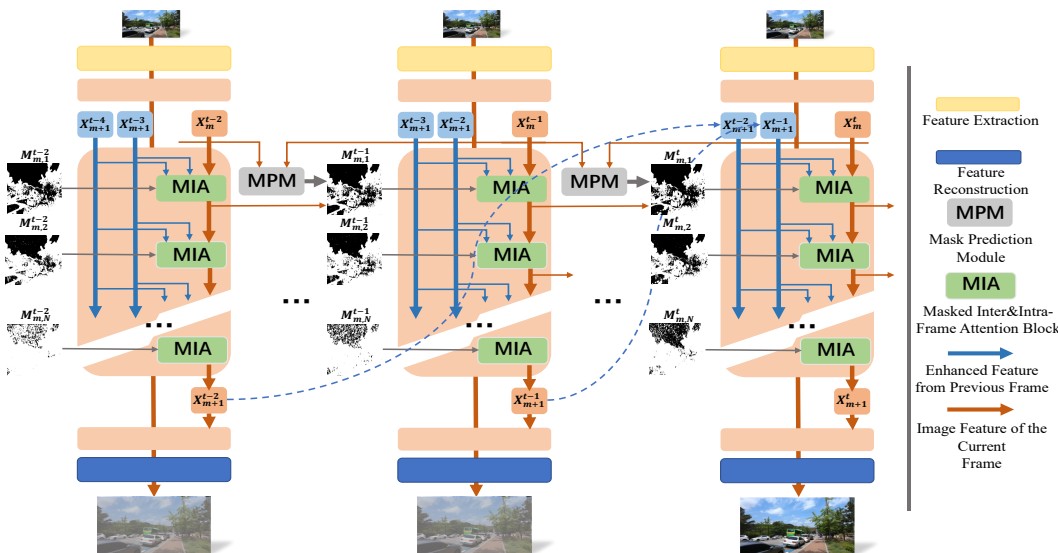

Figure 1: **The overall architecture of MIA-VSR.** Our MIA model is based on the bi-directional second-order grid propagation framework Chan et al. (2022). We develop a feature-level masked processing framework which leverage temporal continuity to reduce redundant computations and propose a masked intra-frame and inter-frame block to make more rational use of previous enhanced features to support the feature enhancement of the current frame. More details of our proposed MIA-VSR can be found in Sec 3.

used to align adjaent frames. We improve the efficiency of existing works by reducing redundant computations in the recurrent feature refinement part.

Generally, the recurrent feature refinement part comprises **M** feature propagation modules and each feature propagation module is consist of **N** cascaded processing blocks. The feature propagation module takes the enhanced outputs from previous frames as well as image feature of the current frame as input. For the $t$-th frame, let's denote the input feature for the $m$-th feature propagation module as $X_m^t$. The feature propagation module takes $X_m^t$, $X_{m+1}^{t-1}$ and $X_{m+1}^{t-2}$ as inputs to calculate enhanced feature:

$$X_{m+1}^t = \texttt{FPM}(X_m^t, X_{m+1}^{t-1}, X_{m+1}^{t-2}), \tag{1}$$

where $X_{m+1}^t$ is the enhanced output of the current frame, $X_{m+1}^{t-1}$ and $X_{m+1}^{t-2}$ are enhanced feature from the past two frames. To be more specific, in each feature propagation module, cascaded processing blocks utilize outputs from the previous frames to enhance input features. In comparison to the previous methods which directly inherit SwinTransformer block to process concatenated features, we propose a tailored intra&inter frame attention block (IIAB) to more efficiently enhance $X_m^t$ with the help of $X_{m+1}^{t-1}$ and $X_{m+1}^{t-2}$:

$$X_{m,n+1}^t = \texttt{IIAB}(X_{m,n}^t, X_{m+1}^{t-1}, X_{m+1}^{t-2}), \tag{2}$$

where $X_{m,n}^t$ is the input to the $n$-th IIAB in the $m$-th feature propagation module, $X_{m,0}^t = X_m^t$ and $X_{m,N}^t = X_{m+1}^t$ are the input and output of the whole FPM, respectively. More details of the proposed IIAB will be introduced in the following subsection 3.2.

In order to reduce redundant computations according to temporal continuity, we further develop an adaptive mask predicting module to generate block-wise masks $M_{m,n}^t$, with which we could directly utilize the outputs from past frame and selectively skip unimportant computation:

$$\hat{X}_{m,n}^t = M_{m,n}^t \odot X_{m,n}^t + (1 - M_{m,n}^t) \odot \hat{X}_{m,n}^{t-1}, \tag{3}$$

where $\odot$ is the point-wise multiplication operation. More details of our mask predicting module will be presented in section 3.3.

## 3.2 INTER&INTRA-FRAME ATTENTION BLOCK

As introduced in the previous section, the cascaded processing blocks play a key role in extracting supplementary information from previous frames to enhancing SR features. To achieve this goal, the previous method (Shi et al., 2022) simply adopt multi-head self-attention block with shifted local windows in Swin Transformer to process concatenated hidden states (i.e. enhanced features of previous frames) and the current input feature. While, in this paper, we take the respective role of previously enhanced feature and the current feature into consideration and propose a intra&inter frame attention block (IIAB) for efficient VSR.

Concretely, we think the enhanced features from previous frames $X_{m+1}^{t-1}$ and $X_{m+1}^{t-2}$ should only used for providing supplementary information and do not need to be further enhanced. Therefore, we only utilize feature of the current frame to generate Query Tokens, and adopt enhanced features from the previous frames as well as feature of the current frame to generate Key and Value Tokens:

$$Q_{m,n}^t = X_{m,n}^t W_{m,n}^Q, \quad \left\{ \begin{array}{l} K_{m,n}^{t,intra} = X_{m,n}^t W_{m,n}^K, \\ K_{m,n}^{t,inter} = \left[ X_m^{t-1}; X_m^{t-2} \right] W_{m,n}^K, \end{array} \right. \quad \left\{ \begin{array}{l} V_{m,n}^{t,intra} = X_{m,n}^t W_{m,n}^V, \\ V_{m,n}^{t,inter} = \left[ X_m^{t-1}; X_m^{t-2} \right] W_{m,n}^V, \end{array} \right. \tag{4}$$

where $W_{m,n}^Q$, $W_{m,n}^K$ and $W_{m,n}^V$ are the respective projection matrices; $Q_{m,n}^t$ represents the Query Tokens generated from the current input feature; $K_{m,n}^{t,intra}$ and $K_{m,n}^{t,inter}$ are the intra-frame and inter-frame Key Tokens and $V_{m,n}^{t,intra}$ and $V_{m,n}^{t,inter}$ are the intra-frame and inter-frame Value Tokens. IIAB jointly calculates the attention map between query token and intra&inter-frame keys to generate the updated feature:

$$\mathrm{IIAB_{Attention}} = \mathrm{SoftMax}(Q_{m,n}^t {K_{m,n}^t}^T / \sqrt{d} + B) V_{m,n}^t, \tag{5}$$

where $K_{m,n}^t = [K_{m,n}^{t,inter}; K_{m,n}^{t,intra}]$ and $V_{m,n}^t = [V_{m,n}^{t,inter}; V_{m,n}^{t,intra}]$ are the concatenated intra&inter-frame tokens; $d$ is the channel dimension of the Token and $B$ is the learnable relative positional encoding. It should be noted that in Eq.4, all the intermediate inter-frame tokes $\{V_{m,n}^{t,inter}; K_{m,n}^{t,inter}\}_{n=1,...,N}$ are generated from the same enhanced features $\left[ x_m^{t-1}; x_m^{t-2} \right]$; which means we only leverage mature enhanced features from previous frames to provide supplementary information and do not need to utilize the time consuming self-attention mechanism to jointly update the current feature and previous features. An illustration of our proposed inter-frame and intra-frame attention (IIAB) block is presented in Fig. 4.2. In addition to the attention block, our transformer layer also utilize LayerNorm and FFN layers, which have been commonly utilized in other Transformer-based architectures (Liang et al., 2022b; Shi et al., 2022).

In our implementation, we also adopt the Shift-window strategy to process the input LR frames and the above self-attention calculation is conduct in $W \times W$ non-overlapping windows. Denote our channel number by $C$, $Q_{m,n}^t$, $K_{m,n}^t$ and $V_{m,n}^t$ in Eq. 5 are with sizes of $W^2 \times C$, $3W^2 \times C$ and $3W^2 \times C$, respectively. Therefore, Eq. 5 only suffers $1/3$ computation in the self-attention calculation procedure in comparison with the joint self-attention processing strategy. Moreover, as we will validate in the ablation study section, our strategy not only avoids unnecessary computation in jointly updating previous features, but also provides better features for the final VSR task.

## 3.3 ADAPTIVE MASKED PROCESSING IN THE RECURRENT VSR TRANSFORMER

In this subsection, we present details of how we generate block-wise masks $M_{m,n}^t$ in Eq. 3 to further reduce unimportant computations. Habibian et al. (2021) firstly proposed a skip convolution mechanism to reduce redundant computations based on differences between pixel values in adjacent frames. However, for the low-level VSR task, skipping the whole computation process of certain region according to intensity differences between adjacent input frames will inevitably affects the accuracy of the model. In this paper, we leverage the temporal continuity to reduce computation in a subtler way: exploiting feature differences between adjacent frames to select less important computations for each block.

Since features of different stages have various contributions to the final VSR result, using feature differences between adjacent frames to generate binary masks through an uniform threshold is non-trivial. Therefore, we propose an adaptive masked processing scheme which jointly trains tiny mask predicting networks with VSR feature enhancement blocks. To be more specific, in each stage of

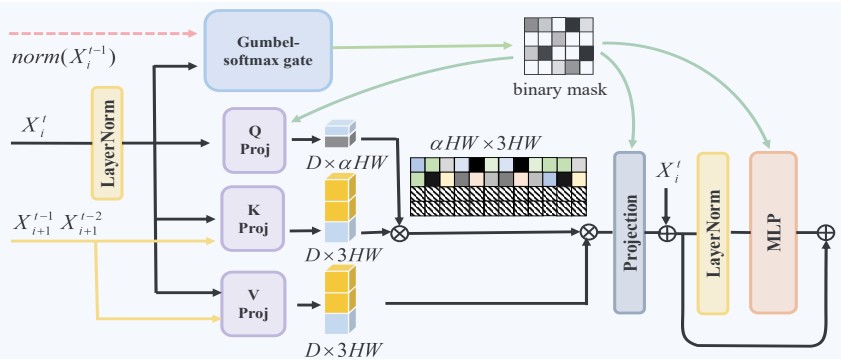

Figure 2: **Illustration of the inter&intra-frame attention block with adaptive masked processing strategy.** The adaptive masked processing strategy in the IIAB block acts in the `Attention` module's linear layer which produce the Quary, the projection layer and the linear layer in the `FFN` module during inference to reduce temporal and sptical redundancy calculations.

processing, the mask predicting network takes the difference between normalized features as input:

$$\Delta X_{m,n}^{t-1 \rightarrow t} = \|\texttt{norm}(X_{m,n}^t) - \texttt{norm}(X_{m,n}^{t-1})\|_1. \tag{6}$$

Then, we employ a Gumbel-softmax gate to sample masking features:

$$
\begin{cases}
\texttt{Mask}(\Delta X_{m,n}^{t-1 \rightarrow t})) = \dfrac{\exp\left((\log(\pi_1) + g_1)/\tau\right)}{\sum_{i=1}^{2} \exp\left((\log(\pi_i) + g_i)/\tau\right)}, \\
\pi_1 = \texttt{Sigmoid}(f(\Delta X_{m,n}^{t-1 \rightarrow t})), \pi_2 = 1 - \texttt{Sigmoid}(f(\Delta X_{m,n}^{t-1 \rightarrow t})),
\end{cases}
\tag{7}
$$

where $f(\cdot)$ is a $1 \times 1$ convolution layer to weighted sum feature differences from different channels; $\pi_1$ could be interpreted as the probability of whether a position should be preserved; and $g_1$, $g_2$ are noise samples drawn from a Gumbel distribution, $\tau$ is the temperature coefficient and we set it as 2/3 in all of our experiments. With the masking feature $\texttt{Mask}(\Delta X_{m,n}^{t-1 \rightarrow t})$, we could directly set a threshold value to generate the binary mask:

$$
M_{m,n}^t = \begin{cases}
1, & if \quad \texttt{Mask}(\Delta X_{m,n}^{t-1 \rightarrow t}) > 0.5; \\
0, & else.
\end{cases}
\tag{8}
$$

The above Gumbel-softmax trick enables us to train mask predicting networks jointly with the VSR network, for learning diverse masking criterion for different layers from the training data. In the inference phase, we do not add Gumbel noise to generate the masking feature and directly generate binary masks with $f(\Delta X_{m,n}^{t-1 \rightarrow t})$.

We apply the same mask to save computations in the $\texttt{IIAB}_{\texttt{Attention}}$ part and the following FFNs in the feature projection part. The above masked processing allows us to generate less Query tokens in the $\texttt{IIAB}_{\texttt{Attention}}$ part and skip projections in the feature projection part. Let's denote $\alpha \in [0, 1]$ as the percentage of non-zeros in $M_{m,n}^t$, our Masked Intra&Inter-frame Attention Block is able to reduce the computational complexity of each IIAB from $9H*W*C^2 + 4H^2W^2$ to $(6 + 3\alpha)H*W*C^2 + (3\alpha + 1)n^2$. The saved computations overwhelm the extra computations introduced by the tiny mask predicting networks, which only use one layer of $1 \times 1$ convolution to reduce the channel number of feature difference tensor from $C$ to 1.

### 3.4 TRAINING OBJECTIVES

We train our network in a supervised manner. Following recent state-of-the-art approaches, we utilize the Charbonnier loss (Charbonnier et al., 1994) $\mathcal{L}_{sr} = \sqrt{\| \hat{I}^{HQ} - I^{HQ} \|^2 + \varepsilon^2}$ between estimated HR image $\hat{I}^{HQ}$ and ground truth image $I^{HQ}$ to train our network; where $\epsilon$ is a constant and we set it as $10^{-3}$ in all of our experiments. Moreover, In order to push our mask predicting networks to mask out more positions, we apply a $\ell_1$ loss on the masking features:

$$\mathcal{L}_{mask} = \frac{1}{MNZ} \sum_{m=1}^{M} \sum_{n=1}^{N} \|\texttt{Mask}(\Delta X_{m,n}^{t-1 \rightarrow t}))\|_1, \tag{9}$$

where $Z$ is the number of pixels for each masking feature. Our network is trained by a combination of $\mathcal{L}_{sr}$ and $\mathcal{L}_{mask}$:

$$\mathcal{L} = \mathcal{L}_{sr} + \lambda\mathcal{L}_{mask}, \tag{10}$$

$\lambda$ is a factor for masking ratio adjustment. More details can be found in our experimental section.

## 4 EXPERIMENTS

### 4.1 EXPERIMENTAL SETTINGS

Following the experimental settings of recently proposed VSR methods (Shi et al., 2022; Liang et al., 2022b;a; Chan et al., 2022; Cao et al., 2021), we evaluate our MIA-VSR model on the REDS (Nah et al., 2019), Vimeo90K (Xue et al., 2019) and the commonly used Vid4 (Liu & Sun, 2013) datasets. We train two models on the REDS dateset and the Vimeo90K dataset, respectively. The model trained on the REDS dataset is used for evaluating the REDS testing data and the Viemo90K model is used for evaluating the Vimeo90K and the Vid4 testing data. We implement our model with Pytorch and train our models with RTX 4090s GPUs. The respective hyper-parameters used for ablation study and comparison with state-of-the-art methods will be introduced in the following subsections.

### 4.2 ABLATION STUDY

**The effectiveness of IIAB** In order to show the advantages of the proposed IIAB, we firstly compare the proposed intra&inter-frame attention block (IIAB) with the multi-frame self-attention block (MFSAB) which was adopted in PSRT (Shi et al., 2022). We use 6 IIAB or MFSAB blocks to build feature propagation modules and instantialize VSR models with 4 feature propagation modules.

Table 1: Ablation studies on the processing blocks and the proposed adaptive masking strategy. More details can be found in our ablation study section.

| Model | $\lambda$ | Params (M) | REDS4 PSNR | SSIM | FLOPs(G) |
|---|---|---|---|---|---|
| MFSAB-VSR | - | 6.41 | 31.03 | 0.8965 | 871.59 |
| IIAB-VSR | - | 6.34 | 31.12 | 0.8979 | 518.92 |
| IIAB-VSR + HM | - | 6.34 | 29.83 | 0.8390 | 420.53 |
| MIA-VSR | 1e-4 | 6.35 | 31.11 | 0.8978 | 506.28 |
| MIA-VSR | 3e-4 | 6.35 | 31.07 | 0.8972 | 469.78 |
| MIA-VSR | 5e-4 | 6.35 | 31.01 | 0.8966 | 442.32 |
| MIA-VSR | 1e-3 | 6.35 | 30.76 | 0.8773 | 426.84 |

The channel number, window size and the head number for the two models are set as 120, 8 and 6, respectively. We denote the two VSR models as IIA-VSR, MFSA-VSR, and train the two models with 8 frame training data for 300K iterations. The super-resolution results by the two models are shown in Table 3. The number of parameters and computational consumption (FLOPs) by the two models are also reported for reference. The number of FLOPs is calculated on the REDS dataset, which super resolve $180 \times 320$ video sequences to a resolution of $720 \times 1280$, we report the average FLOPs for each frame of processing. As shown in Table 1, it is clear that the proposed IIAB could generate better VSR results with less computational consumption than the MFSAB block.

**Masked processing strategy** In this part, we validate the effectiveness of the proposed masked processing strategy. Our basic experimental settings are the same as our experiment for validating IIAB. We firstly present experimental results to test if we could good experimental results by setting an uniform threshold to generate masks based on feature differences. We denote the model as IIAB-VSR + HM and set the threshold value as 0.2, where HM is abbreviation for Handcraft Mask. As can be found in Table 1, adopting an uniform threshold for different layers will lead to a

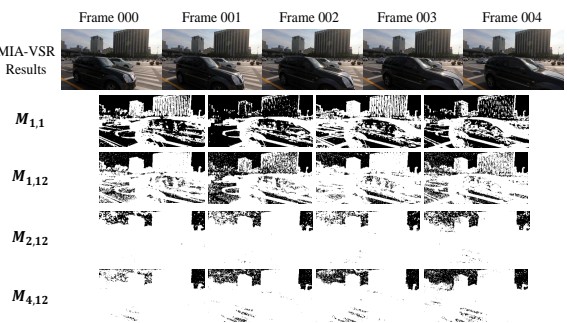

Figure 3: Visualization of predicted masks for a sequence in the REDS dataset.

Table 2: Quantitative comparison (PSNR/SSIM) on the REDS4(Nah et al., 2019), Vimeo90K-T(Xue et al., 2019) and Vid4(Liu & Sun, 2013) dataset for 4× video super-resolution task. For each group of experiments, we color the best and second best performance with red and blue, respectively.

| Method | Frames REDS/Vimeo | REDS4 | | | Vimeo-90K-T | | | Vid4 | | |
|---|---|---|---|---|---|---|---|---|---|---|
| | | PSNR | SSIM | FLOPs | PSNR | SSIM | FLOPs | PSNR | SSIM | FLOPs |
| TOFlow(Xue et al., 2019) | 5/7 | 27.98 | 0.7990 | - | 33.08 | 0.9054 | - | 25.89 | 0.7651 | - |
| EDVR(Wang et al., 2019) | 5/7 | 31.09 | 0.8800 | 2.95 | 37.61 | 0.9489 | 0.367 | 27.35 | 0.8264 | 1.197 |
| MuCAN(Li et al., 2020) | 5/7 | 30.88 | 0.8750 | 1.07 | 37.32 | 0.9465 | 0.135 | - | - | - |
| VSR-T(Cao et al., 2021) | 5/7 | 31.19 | 0.8815 | - | 37.71 | 0.9494 | - | 27.36 | 0.8258 | - |
| PSRT-sliding(Shi et al., 2022) | 5/- | 31.32 | 0.8834 | 1.66 | - | - | - | - | - | - |
| VRT(Liang et al., 2022a) | 6/- | 31.60 | 0.8888 | 1.37 | - | - | - | - | - | - |
| PSRT-recurrent(Shi et al., 2022) | 6/- | 31.88 | 0.8964 | 2.39 | - | - | - | - | - | - |
| MIA-VSR(ours) | 6/- | 32.01 | 0.8997 | 1.59 | - | - | - | - | - | - |
| BasicVSR(Chan et al., 2020) | 15/14 | 31.42 | 0.8909 | 0.33 | 37.18 | 0.9450 | 0.041 | 27.24 | 0.8251 | 0.134 |
| IconVSR(Chan et al., 2020) | 15/14 | 31.67 | 0.8948 | 0.51 | 37.47 | 0.9476 | 0.063 | 27.39 | 0.8279 | 0.207 |
| BasicVSR++(Chan et al., 2022) | 30/14 | 32.39 | 0.9069 | 0.39 | 37.79 | 0.9500 | 0.049 | 27.79 | 0.8400 | 0.158 |
| VRT(Liang et al., 2022a) | 16/7 | 32.19 | 0.9006 | 1.37 | 38.20 | 0.9530 | 0.170 | 27.93 | 0.8425 | 0.556 |
| RVRT(Liang et al., 2022b) | 16/14 | 32.75 | 0.9113 | 2.21 | 38.15 | 0.9527 | 0.275 | 27.99 | 0.8462 | 0.913 |
| PSRT-recurrent(Shi et al., 2022) | 16/14 | 32.72 | 0.9106 | 2.39 | 38.27 | 0.9536 | 0.297 | 28.07 | 0.8485 | 0.970 |
| MIA-VSR(ours) | 16/14 | 32.78 | 0.9220 | 1.61 | 38.22 | 0.9532 | 0.204 | 28.20 | 0.8507 | 0.624 |

significant accuracy drop of the VSR model. Then, we validate the effectiveness of our proposed adaptive masking strategy and analyze the effects of different $\lambda$ values in Eq. 10. Concretely, we set $\lambda$ as $1e-4$, $3e-4$, $5e-4$ and $1e-3$ and train four different models. Generally, setting a larger weight for the sparsity loss could push the network to mask our more computations but also results in less accurate VSR results. By setting $\lambda$ as $5e-4$, we could further save about 20% computations from the highly efficient IIAB-VSR model without significant performance drop. The much better trade-off between accuracy and computation by our MIA-VSR model over IIA-VSR + Handcraft Mask clearly validated the superiority of our adaptive mask predicting network. Some visual examples of the masks generated in our MIA-VSR model can be found in Fig. 4.2. The network tend to skip a large portion of computations in the early stage and could mask less positions for deeper layers.

## 4.3 COMPARISON WITH STATE-OF-THE-ART VSR METHODS

In this subsection, we compare the proposed MIA-VSR model with state-of-the-art VSR methods. We compare the proposed MIA-VSR with representative sliding-window based methods TOFlow(Xue et al., 2019), EDVR (Wang et al., 2019), MuCAN (Li et al., 2020), VSR-T (Cao et al., 2021), VRT (Liang et al., 2022a), RVRT (Liang et al., 2022b) and representative recurrent based methods BasicVSR (Chan et al., 2020), BasicVSR++ (Chan et al., 2022), PSRT-recurrent (Shi et al., 2022); among which VRT, RVRT and PSRT-recurrent are Transformer based approaches and the other approaches are CNN based models.

We instantiate our MIA-VSR model with 4 feature propagation modules and each feature propagation module contains 24 MIIA blocks. Among them, we set the interval of skip connections to [6,6,6,6]. The spatial window size, head size and channel size are set to $8 \times 8$, 6 and 120 accordingly. The number of parameters of our model is on par with recent state-of-the-art methods RVRT (Liang et al., 2022b) and PSRT-recurrent (Shi et al., 2022). We follow the experimental setting of (Liang et al., 2022b; Shi et al., 2022; Liang et al., 2022a) and train VSR model with short sequences (6 frames from the REDS dataset) and long sequences (16/14 frames from the REDS dataset and Vimeo-90K dataset). Generally, models trained with longer sequences often lead to better VSR results.

The VSR results by different results are shown in Table 2. As the number of FLOPs(T) for our MIA-VSR model will be affected by the content of video sequences, we therefore report the average per frame FLOPs on different datasets. In terms of the results trained with short sequences, the proposed MIA-VSR outperform the compared methods by a large margin. Our model improves the PSNR of state-of-the-art PSRT-recurrent model by 0.13 dB with a reduction in the number of FLOPs of almost 40%. As for the models trained with longer training sequences, our MIA-VSR still achieves a better trade-off between VSR accuracy and efficiency over recent state-of-the-art approaches. With more than 40% less computations over the RVRT and PSRT-recurrent approaches, our model achieved the best VSR results on the REDS and Vid4 datasets and the second best results on the Vimeo-90K dataset. Some visual results by different VSR results can be found in Fig. 4, our

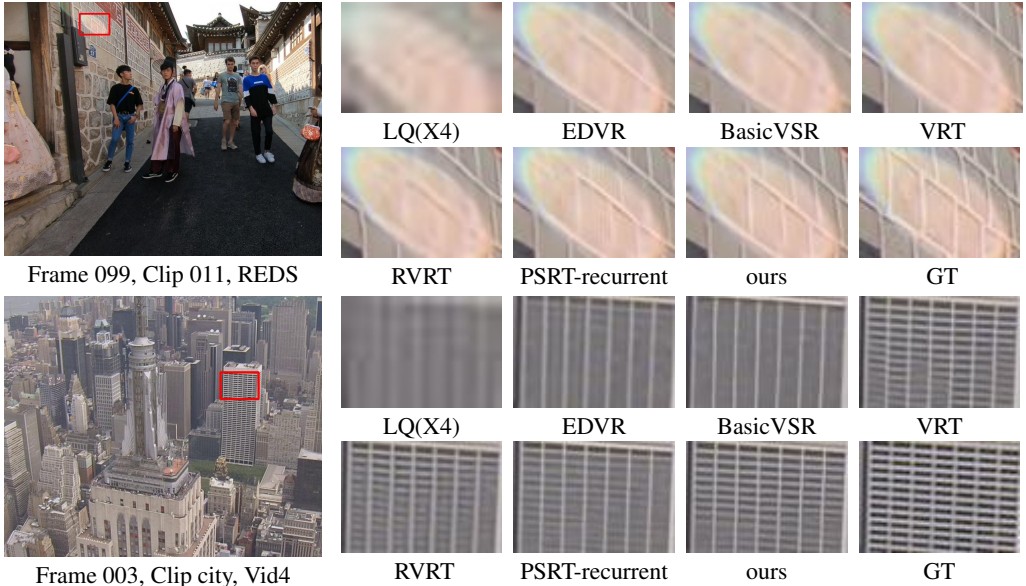

Figure 4: Visual comparison for $4\times$ VSR on REDS4 dataset and Vid4 dataset.

Table 3: Comparison of model size, testing memory and complexity of different VSR models on the REDS dataset.

| Model | Params(M) | FLOPs(T) | Memory(M) | Runtime(ms) | PSNR(dB) |
|---|---|---|---|---|---|
| BasicVSR++ | 7.3 | 0.39 | 223 | 92 | 32.39 |
| VRT | 35.6 | 1.37 | 2149 | 888 | 32.19 |
| RVRT | 10.8 | 2.21 | 1056 | 473* | 32.75 |
| PSRT | 13.4 | 2.39 | 190 | 1041 | 32.72 |
| MIA-VSR | 16.5 | 1.61 | 206 | 822 | 32.78 |

MIA-VSR method is able to recover more natural and sharp textures from the input LR sequences. More visual results can be found in our Appendix.

### 4.4 COMPLEXITY AND MEMORY ANALYSIS

In this part, we compare the complexity and memory footprint of different methods. In Table 3, we report the number of parameters, the peak GPU memory consumption, the number of FLOPs, the Runtime and the PSNR by different VSR methods. Generally, the CNN based BasicVSR++ approach has less complexity and memory requirement over the Transformer based approaches. But its VSR results are not as good as Transformer based methods. In comparison with other Transformer based VSR methods, our MIA-VSR method has similar number of parameters and require much less number of FLOPs for processing the video sequence. In addition, the peak GPU memory consumption, which is critical factor for deploying model on terminal Equipment, by our model is much less than the VRT and RVRT approaches. As for the runtime, our model is not as fast as RVRT, because the authors of RVRT have implemented the key components of RVRT with customized CUDA kernels. As the acceleration and optimization of Transformers are still to be studied, there is room for further optimization of the runtime of our method by our relatively small FLOPs.

### 5 CONCLUSION

In this paper, we proposed a novel Transformer-based recurrent video super-resolution model. We proposed a masked processing frame to leverage the temporal continuity between adjacent frames to save computations for video super-resolution model. An Intra-frame and Inter-frame attention block is proposed to make better use of previously enhanced features to provide supplementary information; and an adaptive mask predicting module is developed to generate blocl-wise masks for each stage of processing. We evaluated our MIA-VSR model on various benchmark datasets. Our model is able to achieve state-of-the-art video super-resolution results with less computational resources.

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

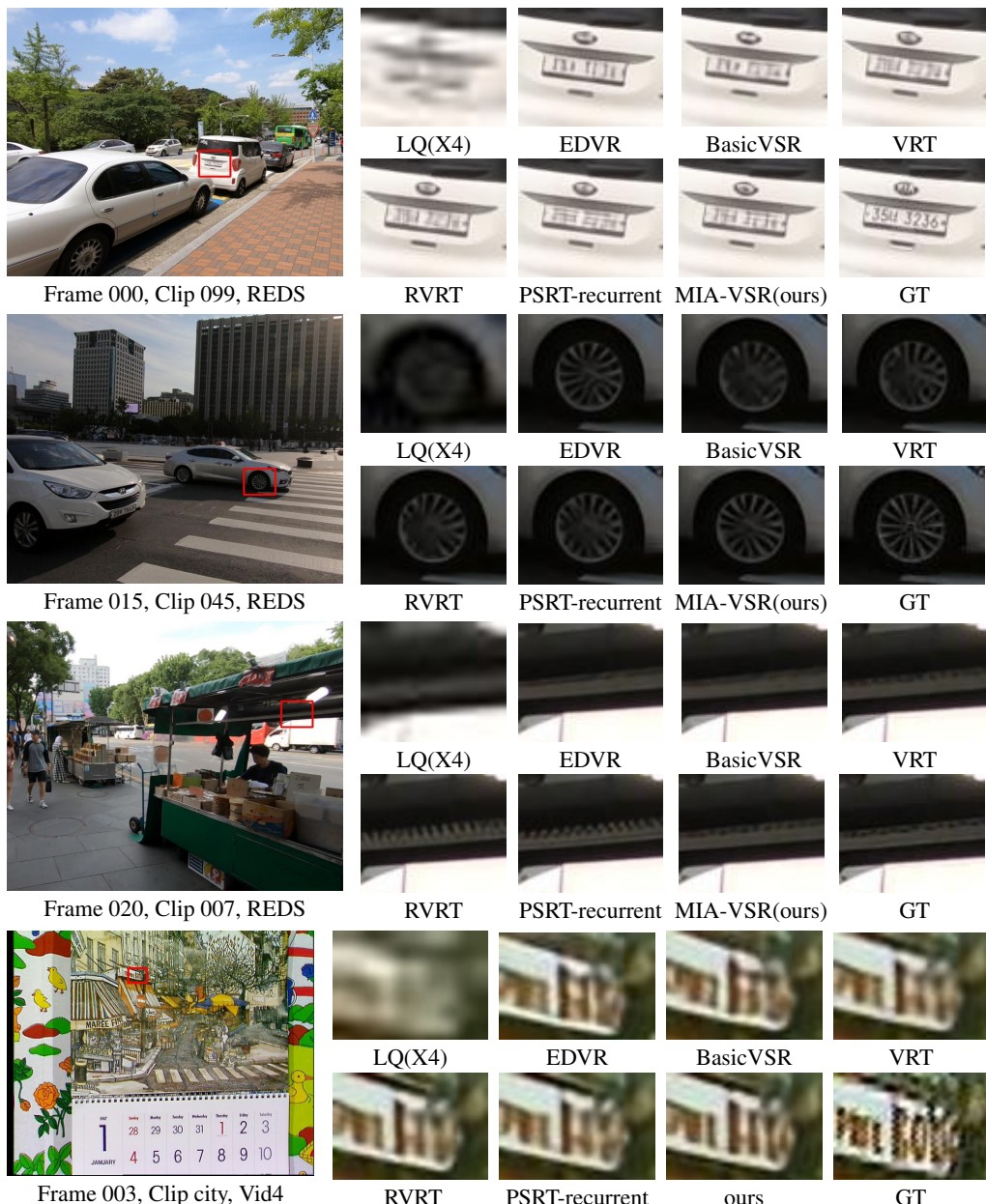

Figure 5: Visual comparison for $4\times$ VSR on REDS4 dataset and Vid4 dataset.

## A  VISUAL RESULTS

We show more visual comparisons between the existing VSR methods and the proposed VSR Transformer with masked inter&intra-frame attention. We use 16 frames to train on the REDS dataset and 14 on the Vimeo-90K dataset. Figure 5 shows the visual results. It can be seen that, in addition to its quantization improvement, the proposed method can generate visually pleasing images with sharp edges and fine details, such as horizontal bar patterns of buildings and numbers on license plates. In contrast, existing methods suffer from texture distortion or loss of detail in these scenes.

