# OpenReview forum: "Video Super-Resolution Transformer with Masked Inter&Intra-Frame Attention"
_ICLR.cc/2024/Conference — ICLR 2024 Conference Withdrawn Submission_

### Official Review · Reviewer_FxaB · 2023-10-28

**Soundness:** 2 fair
**Presentation:** 2 fair
**Contribution:** 2 fair
**Rating:** 3
**Confidence:** 4

**Summary:**

This paper proposes a Transformer-based recurrent video super-resolution model, termed MIA-VSR.
The aim is to reduce the redundant computation in VSR model.
To achieve this goal, they propose two complements: An Intra-frame and Inter-frame attention block (IIAB or MIA) and an adaptive mask predicting module (MPM).
MIA aims to provide supplementary information from the previous enhanced features (temporal information).
MPM  aims to generate block-wise masks to reduce the computation.
Experiments show that MIA-VSR achieves good results on several datasets.

**Strengths:**

1. In the field of video super-resolution, redundant computation is very worthy of study. The problem explored in this paper is meaningful, and the calculation of feature reduction through mask strategy sounds reasonable.
2. The key idea is simple and easy to understand. From the experimental results, the method in this paper seems to be effective.

**Weaknesses:**

1. The core problem to be solved in this paper is the redundant computation in VSR, so the MASK strategy is proposed to reduce the calculation of unimportant features. While designing attention mechanisms to take advantage of timing information has been discussed in many previous works, the second contribution of this article （MIA）does not seem to differ from existing attention mechanisms.
In other words, directly calculating the attention mechanism through the output features after the mask strategy is also computation-intensive. If the author claims the contribution of its attention mechanism, it should be contrasted with this baseline.
2. In terms of reducing redundant computation through the mask strategy, the authors should discuss how it differs from other approaches such as Token Merging and TTVSR. TTVSR also reduces the computation by limiting the attention mechanism to the trajectory of optical flow by calculating the temporal relationship of optical flow. Authors should cite and discuss the differences.
3. Experiments. MASK has a binary ratio, and the author should perform ablation experiments on it, including the FLOPs, not just choose 0.5.  Many new VSR methods are not referenced and compared, such as TTVSR, FTVSR.
4. Writing. The second paragraph of the intro is written like related work. The figures in this paper are messy and not easy to understand, e.g. Fig 1, so many arrows are misunderstood. For example, why two arrows refer to mask M? Is it the output of MPM?
5. The author claimed that aims to solve the problem of heavy computational burden. According to Tab 9, compared with basicvsr++, 7.3M/92ms/32.39dB, MIA-VSR achieves 16.5M/822ms/32.78dB. The results of this experiment do not show its advantages.
To sum up, I think the innovation and contribution of this paper are not obvious enough. Writing and experimentation are not enough. This paper is not sufficient for acceptance by ICLR.

**Questions:**

see weakness

---

### Official Review · Reviewer_FTQC · 2023-10-30

**Soundness:** 2 fair
**Presentation:** 2 fair
**Contribution:** 2 fair
**Rating:** 5
**Confidence:** 4

**Summary:**

The paper proposes a new framework called MIA-VSR for video super-resolution (VSR) tasks. The framework utilizes a feature-level masked processing approach to reduce computational burden and memory footprint, making it more suitable for deployment on constrained devices. The key component of MIA-VSR is the intra-frame and inter-frame attention block, which considers the roles of past features and input features and only uses previously enhanced features to provide supplementary information. Additionally, an adaptive block-wise mask predicting module is developed to skip unimportant computations based on feature similarity between adjacent frames. Ablation studies and comparisons with state-of-the-art VSR methods demonstrate that MIA-VSR improves memory and computation efficiency without sacrificing PSNR accuracy.

**Strengths:**

1. The author tries to accelerate transformer-based VSR from multiple levels, and I think masked processing is reasonable.
2. The  comparative experiments are objective and detailed.

**Weaknesses:**

1. My main concern is that the effect of this method is not significantly improved compared to previous work.

    1.1 From Table 3, choosing transformer for this task does not introduce obvious benefits, especially since CNN can be further compatible with more inference acceleration frameworks. Compared with BasicVSR++, subsequent work uses an order of magnitude higher computational overhead, but it has not made an improvement that I think is worth it. The impression this paper gives me is that it hopes to improve the practicality of this type of method by improving the processing efficiency of VSR. However, the effect of the paper does not seem to achieve this goal. After all, basicVSR++ is already very slow for users.

    1.2 In terms of visual effects comparison, overall there seems to be no significant advantage over PSRT-recurrent.

2. About masked processing

    2.1 I'm worried it's not novel enough. In low-level vision, this kind of blocking processing is not uncommon. Here are a few examples:
* Image SR: Restore Globally, Refine Locally: A Mask-Guided Scheme to Accelerate Super-Resolution Networks
* Background Matting: Real-Time High-Resolution Background Matting

Although this paper does so by considering temporal continuity. Considering the effects shown in Table 1, I think this contribution is insufficient.

**Questions:**

1. How to avoid the blocking effect that mask processing may introduce?
2. I think Figure 1 needs to be redrawn, what is the key message this image is trying to highlight?
3. If generating 720p video requires one second per frame to process, in what scenario do we need video SR?

---

### Official Review · Reviewer_oDvo · 2023-10-31

**Soundness:** 2 fair
**Presentation:** 2 fair
**Contribution:** 2 fair
**Rating:** 3
**Confidence:** 5

**Summary:**

To address the heavy computational burden and large memory footprint in Transformer-based Video Super-resolution (VSR), this paper proposes a masked intra and inter frame attention (MIA-VSR). MIA-VSR uses feature-level temporal continuity between adjacent frames. The experiments demonstrate the effectiveness of the proposed method.

**Strengths:**

1. This paper proposes an intra-frame and inter-frame attention block to enhance SR features, and proposes an adaptive mask predicting module to mask out unimportant regions between adjacent frames.
2. Compared with existing Transformer-based VSR methods, the proposed method has less computational cost and memory footprints.

**Weaknesses:**

1. The novelty of this paper is not clear.
2. The performance is not significant on benchmark datasets. Although the proposed method has less computational cost and memory footprints compared with existing Transformer-based VSR methods, it is still challenging for applications on smartphones (the main issue that the authors highlight to solve.)

**Questions:**

1. The motivations of the paper are to reduce the computational burden and the large memory footprint, and propose a VSR method in smart phones and consumer electronic products. However, the model size is large and not very efficient. For real applications, BasicVSR++ has more advantages than the proposed MIA-VSR. Compared with MIA-VSR, RVRT has a smaller model size, less runtime and comparable PSNR.

2. Some details in Figure 1 are not clear. For example, the inputs of MPM are not clear. How to get x_m^{t-2} in the orange block? What are the blue dashed lines? Why are the output video results poor?

3. The performance is not significant under different metrics. In addition, in Figure 4, it would be better to provide BasicVSR++ results instead of BasicVSR or EDVR.

---

### Official Review · Reviewer_kfYa · 2023-11-01

**Soundness:** 3 good
**Presentation:** 3 good
**Contribution:** 3 good
**Rating:** 6
**Confidence:** 4

**Summary:**

This paper presents a novel Transformer-based video super-resolution model called MIA-VSR (Masked Intra and Inter-frame Attention Video Super-Resolution). The model aims to improve the efficiency of video super-resolution by leveraging temporal continuity between adjacent frames and reducing redundant computations. The key components of MIA-VSR include an intra-frame and inter-frame attention block (IIAB) and an adaptive mask predicting module.

**Strengths:**

1. Improved efficiency: MIA-VSR reduces computational complexity and memory footprint without sacrificing video super-resolution performance.
2. Effective use of temporal information: The model leverages temporal continuity between frames to avoid unnecessary computations and provide better results.
3. Adaptive masking: The adaptive mask predicting module generates block-wise masks to skip unimportant computations, further improving efficiency.

**Weaknesses:**

1. Complexity: The model may be more complex to implement and train compared to simpler video super-resolution methods.
2. Limited applicability: The effectiveness of MIA-VSR may be limited to specific video super-resolution tasks and datasets.
3. Runtime: Although MIA-VSR reduces computational complexity, its runtime may still be slower than some other methods due to the Transformer architecture.

**Questions:**

1. In the comparison with state-of-the-art methods, you mentioned that MIA-VSR achieves better trade-offs between accuracy and efficiency. How does MIA-VSR handle the trade-off between model size and computational efficiency? Can you provide more quantitative analysis or visualizations to support this claim?
2. Can you provide some insights on the design choices for the Intra-frame and Inter-frame Attention Block (IIAB)? How does it differ from other attention mechanisms used in video super-resolution models?